# scDataset: Scalable Data Loading for Deep Learning on Large-Scale Single-Cell Omics

**Davide D'Ascenzo** [1 2]   **Sebastiano Cultrera di Montesano** [3]

## Abstract

Training deep learning models on single-cell datasets with hundreds of millions of cells requires loading data from disk, as these datasets exceed available memory. While random sampling provides the data diversity needed for effective training, it is prohibitively slow due to the random access pattern overhead, whereas sequential streaming achieves high throughput but introduces biases that degrade model performance. We present scDataset, a PyTorch data loader that enables efficient training from on-disk data with seamless integration across diverse storage formats. Our approach combines block sampling and batched fetching to achieve quasi-random sampling that balances I/O efficiency with minibatch diversity. On Tahoe-100M, a dataset of 100 million cells, scDataset achieves more than two orders of magnitude speedup compared to true random sampling while working directly with AnnData files. We provide theoretical bounds on minibatch diversity and empirically show that scDataset matches the performance of true random sampling across multiple classification tasks and model architectures.

## 1. Introduction

Efficient data loading has emerged as a critical bottleneck in training large-scale deep learning models. Modern datasets routinely exceed available memory capacity, with collections spanning billions of samples and terabytes of storage. Since loading entire datasets into memory has become the exception rather than the norm, training requires careful orchestration between two conflicting demands: stochastic gradient descent requires randomized sampling to achieve unbiased gradient estimates and robust model convergence (Bottou, 1999; Keskar et al., 2017), while disk I/O systems are optimized for sequential access patterns and suffer severe performance degradation under random access workloads.

Single-cell omics exemplifies this challenge while introducing domain-specific complications that further amplify the data loading bottleneck. Single-cell RNA sequencing (scRNA-seq) captures gene expression profiles from individual cells, enabling unprecedented insights into cellular heterogeneity, developmental trajectories, and disease mechanisms (Tirosh et al., 2016; Wang & Navin, 2015; Buenrostro et al., 2015). The field has rapidly scaled to datasets containing hundreds of millions of cells. The Human Cell Atlas (Regev et al., 2017) and CellxGene (CZI Cell Science Program et al., 2024) initiatives aim to profile every cell type across diverse tissues, species, and conditions, while recent perturbation screens like Tahoe-100M (Zhang et al., 2025) measure approximately 100 million transcriptomic profiles across 50 cancer cell lines, 380 drugs, and 3 dosages (representing roughly 60,000 experimental conditions with ~2,000 cells sequenced per condition). These datasets are typically stored in the AnnData format (Virshup et al., 2024), which has become the community standard for single-cell analysis. AnnData leverages HDF5 (The HDF Group) to efficiently represent sparse cell-by-gene matrices, supporting both in-memory and on-disk datasets with capabilities for lazy concatenation of sharded collections. This standardization has enabled a rich ecosystem of computational biology tools, most notably within the scverse framework including scanpy (Wolf et al., 2018) for exploratory analysis and scvi-tools (Gayoso et al., 2022) for probabilistic modeling.

While AnnData excels at biological workflows, its design priorities create inherent conflicts with deep learning requirements. Sparse matrix representations that enable compact storage (~300GB for Tahoe-100M) must be converted to dense formats for neural network consumption, expanding memory footprints by orders of magnitude (>1TB for Tahoe-100M). The HDF5 backend is optimized for contiguous sequential reads, not the random access patterns required for shuffled minibatch sampling. Perhaps most

[1]Department of Computer Science, University of Milan, Milan, Italy [2]Department of Control and Computer Engineering, Politecnico di Torino, Torino, Italy [3]Eric and Wendy Schmidt Center, Broad Institute of MIT and Harvard, Cambridge, MA, USA. Correspondence to: Davide D'Ascenzo <davide.dascenzo@unimi.it>, Sebastiano Cultrera di Montesano <scultrer@broadinstitute.org>.

*Proceedings of the 43rd International Conference on Machine Learning*, Seoul, South Korea. PMLR 306, 2026. Copyright 2026 by the author(s).

critically, any departure from the AnnData format breaks compatibility with the established ecosystem, forcing researchers to choose between training efficiency and downstream analysis capabilities. For datasets at the scale of Tahoe-100M stored on disk, these compounding factors make I/O the dominant bottleneck, often rendering deep learning training computationally infeasible without specialized high-memory infrastructure.

The limitations of existing approaches become evident when considering three baseline strategies. Pure random sampling from disk, as implemented in traditional map-style datasets[1], performs one random access per sample: for a typical minibatch of 64 cells, this requires 64 random accesses per iteration. At the scale of Tahoe-100M, this approach yields approximately 20 samples per second, requiring over 58 days to iterate over the full dataset once.[2] Loading entire datasets into memory would eliminate I/O overhead but is prohibitively expensive: Tahoe-100M occupies $\sim$300GB in its sparse compressed AnnData format and would require more than 1TB when decompressed in memory or converted to dense representations, far exceeding typical RAM budgets and precluding work with even larger forthcoming datasets. Streaming data sequentially from disk achieves high I/O throughput but introduces severe training biases. For instance, Tahoe-100M is organized by experimental plates containing approximately 7 million cells each: sequential streaming would expose models to millions of cells from similar experimental conditions consecutively, potentially leading to catastrophic forgetting and model collapse (Kirkpatrick et al., 2017). Format conversion (e.g., to Parquet as used by HuggingFace Datasets), combined with pre-shuffling, can offer faster contiguous read patterns, but incurs substantial costs. Converting Tahoe-100M from its native AnnData format substantially increases storage requirements, and the conversion process itself is computationally expensive for large datasets. More critically, this approach sacrifices flexibility: common workflow modifications such as changing train–test splits, adding new datasets, adjusting data filtering, or experimenting with different sampling strategies require repeating the entire conversion and pre-shuffling process. In addition, converted data must be transformed back to AnnData for downstream analysis with scvi-tools and other ecosystem tools. Intermediate approaches such as WebDataset (Aizman et al., 2019) and Ray Data (Moritz et al., 2018) implement shuffle buffers that prefetch sequential chunks and yield random subsets from the buffer. However, when the buffer size is small relative to dataset heterogeneity (e.g., a 10K-cell buffer when plates contain $\sim$7 million cells each), the effective sampling

remains highly biased (as we will show in Section 4.4). Increasing buffer size to approach true random sampling would require memory budgets comparable to loading the full dataset.

To reconcile the competing demands of I/O efficiency and minibatch diversity we developed scDataset, a PyTorch IterableDataset that enables fast, randomized, scalable data loading without format conversion or specialized infrastructure. Our approach balances three goals: practical utility as a drop-in solution for existing data collections, theoretical guarantees on minibatch diversity, and performance that enables atlas-scale training on standard hardware. The core innovation lies in trading pure random sampling for quasi-random sampling through two complementary mechanisms: block sampling and batched fetching. Rather than accessing cells individually, scDataset partitions the dataset into contiguous blocks of size $b$ and samples blocks uniformly, reducing random disk accesses from $m$ to $m/b$ per minibatch at the cost of some within-batch correlation. Batched fetching compensates by prefetching $f$ minibatches worth of data in a single operation: the combined buffer is reshuffled in memory and partitioned into $f$ diverse minibatches, each drawing from far more distinct blocks than a single-minibatch fetch would allow. Together, $b$ and $f$ provide explicit control over the throughput–diversity trade-off, which we formalize in Section 3.4.

The block sampling and batched fetching mechanisms are described in detail in Sections 3.1–3.2. We complement this description with theoretical and empirical evidence supporting the proposed design. Specifically, we derive explicit bounds on the expected minibatch entropy as a function of block size and fetch factor (Section 3.4). We then evaluate data-loading performance on the Tahoe-100M dataset and show that scDataset achieves over two orders of magnitude higher throughput than true random sampling from disk while operating directly on AnnData files (Section 4.1). Beyond throughput, we show that scDataset recovers minibatch entropy indistinguishable from true random sampling (Section 4.3). Finally, we evaluate downstream learning performance on multiple classification tasks and show that scDataset matches the accuracy of true random sampling while substantially reducing end-to-end training time (Section 4.4).

Code is available at https://github.com/scDataset/scDataset.

## 2. Related Work

The data loading landscape for single-cell omics has seen several recent developments, each addressing different aspects of the challenge, but none achieving the combination of performance, compatibility, and ease of use that modern

---

[1]https://pytorch.org/docs/stable/data.html#map-style-datasets

[2]All throughput figures in this paper were measured on an NVIDIA DGX Station with 256GB RAM, an Intel Xeon E5-2698 v4 CPU, and 5TB SATA SSD storage.

atlas-scale training demands.

**AnnLoader**. AnnLoader is a PyTorch data loader developed by the AnnData team to enable on-disk sampling from AnnData files. It extends the standard PyTorch `__getitem__` interface to accept both single indices and batches of indices, allowing minibatches to be retrieved through a `batch_sampler` in a single call rather than multiple separate calls. While this batched index retrieval reduces I/O call overhead compared to the naive approach, AnnLoader remains fundamentally constrained by its random access pattern: each minibatch still requires reading from many scattered locations on disk. Our benchmarks on Tahoe-100M show AnnLoader achieves only ~20 samples per second, requiring over 58 days for a single epoch. This throughput is so low that any neural network model consumes data faster than it can be fetched from disk. Furthermore, AnnLoader does not natively support multiprocessing, limiting scalability.[3]

**scDataLoader** (Kalfon et al., 2025). scDataLoader is a PyTorch data loading library tightly coupled to LaminDB[4], a specialized database system for managing biological data collections. Unlike tools that work directly with standard AnnData files, scDataLoader requires datasets to be ingested into LaminDB's infrastructure, introducing an additional dependency layer for data management. While it streamlines preprocessing and annotation workflows for users already committed to that ecosystem, it does not address the fundamental I/O bottlenecks of random access sampling at atlas scale. scDataset can operate directly on LaminDB's MappedCollection objects, allowing it to complement scDataLoader where efficient on-disk sampling is needed.

**AnnBatch** (Gold et al., 2026). AnnBatch is a minibatch loading library co-developed by Lamin Labs and scverse for on-disk AnnData files. The tool converts existing .h5ad files into zarr-backed formats with shuffled, chunked storage layouts optimized for sequential access. While this preprocessing can improve read performance for contiguous access patterns, it requires datasets to be converted into a custom zarr format before training, imposing upfront conversion costs and storage duplication.

**HuggingFace Datasets** (Lhoest et al., 2021). HuggingFace Datasets is a general-purpose framework for dataset management with sophisticated caching, memory mapping, and support for diverse data formats. While powerful for NLP and computer vision workflows, it incurs substantial overhead when applied to single-cell omics: converting Tahoe-100M from its native 300GB AnnData format to Parquet expands storage to ~2TB, and even after conversion, random sampling remains slow due to the inherent disk access patterns.

**BioNeMo-SCDL** (John et al., 2024). BioNeMo-SCDL is NVIDIA's PyTorch-compatible data loader for single-cell foundation model training. The framework converts AnnData files into memory-mapped NumPy arrays stored in a custom format, enabling larger-than-memory datasets to be accessed efficiently. Like other format-conversion approaches, this requires preprocessing datasets before training and duplicates storage, with any subsequent dataset modifications necessitating re-conversion. scDataset can operate directly on BioNeMo's memory-mapped format, enabling users to exploit both the memory-mapping efficiency and scDataset's quasi-random sampling strategies.

**TileDB-SOMA**[5]. TileDB-SOMA provides Python and R APIs for cloud-native access to single-cell data, implementing the SOMA specification (Stack Of Matrices, Annotated) on top of TileDB's array storage format. It powers the CZ CELLxGENE Census, which provides efficient access to a corpus of over 100 million cells. TileDB-SOMA excels at cloud-based data access and larger-than-memory operations with query capabilities for cell and gene metadata. However, its design prioritizes interactive exploration and data retrieval rather than the iterative, epoch-based training patterns central to deep learning. TileDB-SOMA and scDataset address complementary use cases and could in principle be combined for Census-scale training workflows.

**Shuffle Buffer Approaches**. WebDataset (Aizman et al., 2019) and Ray Data (Moritz et al., 2018) implement in-memory shuffle buffers that load sequential chunks and randomly sample from these fixed-size windows. As discussed in Section 1, buffer sizes are constrained by available memory yet must be large relative to dataset heterogeneity to provide sufficient randomness. For example, on Tahoe-100M where experimental plates contain approximately 7 million cells, buffers large enough to span plate-scale diversity would require memory budgets approaching full dataset loading, while smaller buffers (e.g., 10K cells) sampled from limited spatial regions do not mitigate the biases of sequential streaming. Both approaches also require format conversion from AnnData, breaking scverse ecosystem compatibility.

## 3. Methods

We present scDataset, a PyTorch `IterableDataset` designed to enable efficient deep learning training on large-scale datasets stored on disk. Our approach addresses the fundamental I/O bottleneck through a quasi-random sampling strategy that balances the randomness requirements of stochastic gradient descent with the sequential access patterns that optimize disk throughput.

---

[3] https://github.com/scverse/anndata/issues/767
[4] https://github.com/laminlabs/lamindb
[5] https://github.com/single-cell-data/TileDB-SOMA

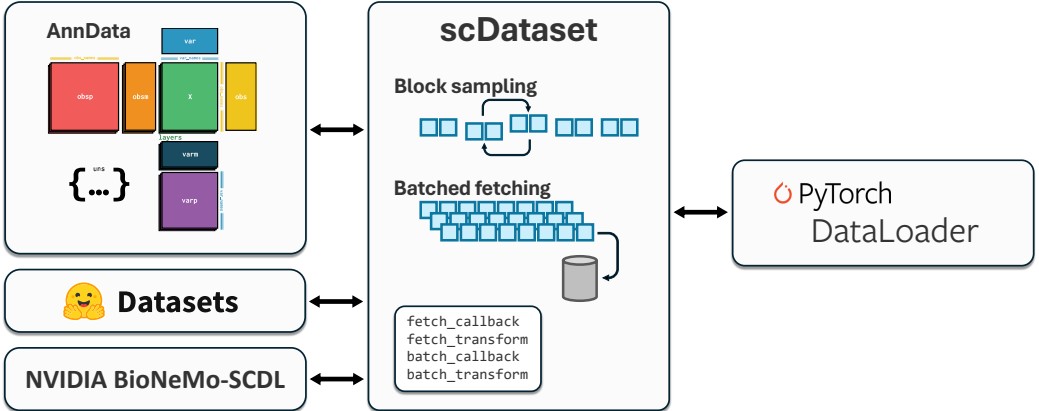

*Figure 1.* scDataset bridges diverse data backends with PyTorch's DataLoader through a modular interface. Data retrieval is managed by a configurable fetch_callback, followed by preprocessing with fetch_transform (e.g., sparse-to-dense conversion). Batches are selected using batch_callback and further processed with batch_transform before being yielded to the training pipeline.

scDataset serves as a flexible interface between diverse data backends and PyTorch's DataLoader (Figure 1). It operates on any indexable data collection, including AnnData objects, HuggingFace Datasets, NumPy arrays, and custom backends, and yields minibatches suitable for neural network training. To achieve high throughput while maintaining sufficient minibatch diversity, scDataset combines block sampling and batched fetching, which we describe next.

### 3.1. Block Sampling

Traditional random sampling for minibatch construction samples each cell independently and uniformly from the dataset. For a minibatch of size $m$, this requires $m$ random disk accesses, one for each cell. At the scale of modern single-cell datasets, each random access incurs significant I/O overhead, making this approach prohibitively slow.

Block sampling addresses this by partitioning the dataset into contiguous blocks of size $b$ and sampling blocks uniformly rather than individual cells. To construct a minibatch of size $m$, we sample $\lceil m/b \rceil$ blocks uniformly at random and read all $b$ cells from each sampled block contiguously. This reduces the number of random disk accesses from $m$ to $\lceil m/b \rceil$.

For example, with $m = 64$ and $b = 16$, traditional random sampling requires 64 separate I/O operations, while block sampling requires only 4 (sampling 4 blocks of 16 cells each). Each operation is followed by a contiguous read of 16 cells, which modern storage systems handle efficiently. The block size $b$ parameterizes the trade-off between I/O efficiency and sampling diversity: larger blocks reduce I/O operations but increase correlation within minibatches, while smaller blocks approach true random sampling at the cost of more disk accesses.

### 3.2. Batched Fetching

Batched fetching further amplifies I/O efficiency by retrieving multiple minibatches worth of data in a single operation. Rather than fetching exactly $m$ cells per iteration, we introduce a fetch factor $f$ and retrieve $m \times f$ cells at once. These cells are stored in an in-memory buffer, reshuffled randomly, and then partitioned into $f$ minibatches for subsequent training iterations.

The key insight is that disk systems can optimize batch requests more effectively than sequential individual requests. Modern HDDs and SSDs can coalesce multiple read requests to minimize head movement (for HDDs) or exploit parallelism in flash arrays (for SSDs). In cloud storage environments, batching also reduces network overhead by issuing a single large request instead of multiple small requests, each of which incurs round-trip latency and HTTP header overhead. By presenting the storage system with $\lceil (m \times f)/b \rceil$ block requests simultaneously, we enable these optimizations.

For instance, with $m = 64$, $b = 16$, and $f = 10$, we sample 40 blocks (640 cells total) in a single batched operation. The 640 cells are then reshuffled in memory (a cheap operation) and partitioned into 10 diverse minibatches. Importantly, without batched fetching, a minibatch of 64 cells would contain contributions from only 4 locations (4 blocks × 16 cells). With $f = 10$, each minibatch draws from up to 40 distinct blocks, dramatically increasing spatial diversity across the dataset.

The fetch factor $f$ controls a three-way trade-off: larger $f$ improves I/O efficiency and increases minibatch diversity (drawing from $m/b \times f$ distinct blocks rather than just $m/b$), but increases memory consumption (storing $m \times f$ cells in

the buffer).[6]

The detailed procedure for block sampling and batched fetching is presented in **Algorithm 1**. We first generate a full array of indices $I$ (Line 1): for a dataset of $n = 10^8$ cells, this requires only ~400 MB since each 32-bit integer occupies 4 bytes. All subsequent operations in Lines 2–5 manipulate these lightweight integer indices in memory, avoiding any disk I/O. We partition $I$ into $k = n/b$ contiguous blocks (Line 2), shuffle the block order uniformly at random (Line 3), and concatenate the shuffled blocks to obtain $I_{\text{shuffled}}$ (Line 4). This block-level shuffling ensures that cells are sampled quasi-randomly across the entire dataset while preserving contiguous access patterns within blocks. We then partition $I_{\text{shuffled}}$ into fetch batches $F_i$ of size $m \cdot f$ (Line 5).

---

**Algorithm 1** Block Sampling with Batched Fetching

**Input** : Dataset size $n$, block size $b$, minibatch size $m$, fetch factor $f$
 (where $n$ is a multiple of $b$ and $m \cdot f$ for simplicity)
**Output** : Sequence of minibatches $\mathcal{M}_0, \mathcal{M}_1, \ldots$
1  Generate full index array: $I = [0, 1, \ldots, n-1]$
2  Split $I$ into $k = \frac{n}{b}$ blocks $[B_0, B_1, \ldots, B_{k-1}]$,
   where $B_i = [i \cdot b, \ldots, (i+1) \cdot b - 1]$
3  Shuffle block order:
   $[B_{\sigma(0)}, \ldots, B_{\sigma(k-1)}] \leftarrow \text{RandomPerm}([B_0, \ldots, B_{k-1}])$
4  Concatenate shuffled blocks:
   $I_{\text{shuffled}} \leftarrow B_{\sigma(0)} \| \ldots \| B_{\sigma(k-1)}$
5  Split $I_{\text{shuffled}}$ into batches $[F_0, \ldots, F_{\frac{n}{m \cdot f}-1}]$ of size $m \cdot f$
6  **for** each $F_i$ **do**
7  | Sort indices in $F_i$ in ascending order
8  | Load data: $\mathcal{F}_i \leftarrow \text{ReadFromDisk}(F_i)$
9  | Shuffle $\mathcal{F}_i$ in memory
10 | Split $\mathcal{F}_i$ into minibatches $\mathcal{M}_0, \ldots, \mathcal{M}_{f-1}$
11 | **for** each $\mathcal{M}_j$ **do**
12 | | **yield** $\mathcal{M}_j$

---

The data loading phase (Lines 6–11) processes each fetch batch $F_i$ independently. We first sort the indices within $F_i$ in ascending order (Line 7), enabling storage backends to coalesce nearby reads and optimize sequential access. We then load the actual data $\mathcal{F}_i$ from disk (Line 8)—note the calligraphic notation $\mathcal{F}_i$ distinguishing loaded data from the integer indices $F_i$. Once in memory, we shuffle $\mathcal{F}_i$ (Line 9), partition the shuffled data into $f$ minibatches $\{\mathcal{M}_0, \ldots, \mathcal{M}_{f-1}\}$ (Line 10), and yield each minibatch to the training loop (Lines 11–12). This design ensures that ex-

---

[6]PyTorch's `DataLoader` prefetch factor controls how many batches each worker preloads to hide scheduling latency and does not affect sample selection or ordering. scDataset's fetch factor $f$ controls the in-memory buffer used for within-buffer shuffling to recover diversity from block-structured storage. The two mechanisms are complementary and operate in tandem.

pensive disk operations occur only at Line 8, while all other operations manipulate lightweight indices or in-memory data.

### 3.3. Implementation Details

scDataset exposes a strategy-based API where users select from predefined sampling strategies: `Streaming` for sequential access (with optional shuffle buffer), `BlockShuffling` for the block sampling described above, `BlockWeightedSampling` for weighted sampling with arbitrary per-cell weights (enabling custom stratification schemes such as upsampling rare cell types or balancing across datasets of unequal size), and `ClassBalancedSampling`, which derives balancing weights automatically from user-provided class labels. Each strategy generates indices that are then processed through the batched fetching pipeline.

Four optional callback hooks decouple data access logic from sampling logic, enabling scDataset to operate on any indexable collection including AnnData, HuggingFace Datasets, and custom backends; see Appendix A for the full pipeline description. Multiprocessing and Distributed Data Parallel (DDP) training are natively supported, with automatic worker-level index partitioning and rank-aware fetch scheduling (Appendix B).

### 3.4. Theoretical Guarantees on Minibatch Diversity

We analyze how block sampling and batched fetching affect minibatch diversity through the lens of label entropy. Our goal is to characterize how the parameters $b$ (block size) and $f$ (fetch factor) control the deviation from true random sampling, and to provide explicit bounds on this deviation.

We assume that each contiguous block of size $b$ is homogeneous with respect to a categorical metadata label (e.g. experimental plate), and that blocks are drawn independently from a categorical distribution $p = (p_1, \ldots, p_K)$ over $K$ labels. This plate-constant assumption reflects the organization of datasets such as Tahoe-100M, where contiguous regions correspond to single experimental conditions.

A minibatch of size $m$ is constructed by sampling $fB$ blocks, where $B = m/b$, forming a buffer of $fm$ cells, uniformly reshuffling this buffer, and selecting $m$ cells. Let $C_k$ denote the number of cells from label $k$ in the minibatch, and define the minibatch entropy

$$H(C) = -\sum_{k=1}^{K} \frac{C_k}{m} \log_2 \frac{C_k}{m}. \tag{1}$$

We analyze the expected entropy $\mathbb{E}[H(C)]$ under this sampling scheme using the classical bias expansion of the plug-in entropy estimator for multinomial samples (Paninski, 2003). Detailed proofs are provided in Appendix C.

When $f \to \infty$, the buffer converges in distribution to the population $p$, and minibatch sampling becomes asymptotically equivalent to drawing $m$ IID cells; the expected entropy approaches $H(p)$ with a bias of order $1/m$ (Theorem C.1). Conversely, when $f = 1$, the effective sample size drops from $m$ to $B = m/b$: only $B$ independent block draws contribute to the minibatch, amplifying the entropy bias by a factor of $b$. In the extreme case $b = m$, the minibatch is a single block and entropy collapses to zero (Theorem C.2). Together, the two regimes yield a sandwich bound for any $f \geq 1$:

**Proposition 3.1** (Minibatch entropy bound). *For any $f \geq 1$,*

$$H(p) - \frac{(K-1)b}{2m \ln 2} \ \leq \ \mathbb{E}[H(C)] \ \leq \ H(p) - \frac{K-1}{2m \ln 2}. \quad (2)$$

We validate these bounds on Tahoe-100M, where the 14 experimental plates have non-uniform sizes ranging from 4.7% to 10.4% of total cells. The empirical plate distribution has entropy $H(p) = 3.78$ bits, slightly below the uniform case ($\log_2 14 = 3.81$ bits). For $m = 64$ and $b = 16$, the bounds give

$$1.43 \ \leq \ \mathbb{E}[H(C)] \ \leq \ 3.63. \quad (3)$$

Empirically, with $f = 1$ (no batched fetching) we measure an entropy of $1.76 \pm 0.33$, close to the lower bound. Increasing to $f = 256$, the empirical entropy rises to $3.61 \pm 0.08$, approaching the upper bound. This confirms that moderate fetch factors suffice to recover near-random minibatch diversity in practice.

As a practical guideline, we recommend $b \leq f/2$. Even on Tahoe-100M, where cells are strictly ordered by plate (a near worst-case scenario for block sampling), Figure 4 shows that $b = f$ already recovers near-random entropy, so $b = f/2$ is a deliberately conservative recommendation.

**Connection to gradient variance.** The entropy analysis admits a direct interpretation in terms of the variance of the minibatch gradient estimator, i.e., how much the gradient changes from one randomly drawn batch to the next. Block sampling is equivalent to cluster sampling (Cochran, 1977): rather than drawing $m$ individual cells, the sampler draws $B = m/b$ contiguous blocks of $b$ cells each. Under the plate-constant assumption, all $b$ cells within a block share the same expected gradient direction, so the minibatch contains only $B$ statistically independent gradient signals. By standard results for cluster sampling (Bottou, 2010), the variance of the gradient estimator scales as $\sigma^2/B = b\sigma^2/m$, larger than the IID estimator $\sigma^2/m$ by a factor of $b$. With fetch factor $f$, the buffer spans $fB = fm/b$ independently sampled blocks; for $f \geq b$, the buffer contains at least $m$ distinct block sources, and after in-buffer reshuffling each minibatch draws from a substantially more diverse pool,

recovering IID-like gradient variance. This argument is a heuristic approximation valid under the plate-constant model; in practice, within-block gradient variation reduces the variance inflation below the worst-case factor $b$. The training dynamics in Section 4.4 provide empirical support for this analysis: streaming training exhibits loss spikes at plate boundaries consistent with the training instability and catastrophic forgetting (Kirkpatrick et al., 2017) predicted when gradients are dominated by the current plate's distribution.

## 4. Results

We evaluated scDataset on Tahoe-100M (Zhang et al., 2025), a single-cell perturbation screen comprising approximately 100 million transcriptomic profiles across 50 cancer cell lines, 380 drugs, and 3 dosages. The dataset measures gene expression (62,710 genes per cell) in individual cells following chemical perturbations, capturing cellular responses across roughly 60,000 experimental conditions with ~2,000 cells sequenced per condition. Tahoe-100M is distributed as 14 AnnData files corresponding to experimental plates, each containing approximately 7 million cells, totaling 314GB on disk in compressed sparse format.

All experiments were conducted on an NVIDIA DGX Station with 256GB RAM, an Intel Xeon E5-2698 v4 CPU, and 5TB SATA SSD storage. We used a fixed minibatch size of 64 throughout all the experiments.

### 4.1. Data Loading Throughput

We measured single-core throughput (samples per second) across a grid of block sizes $b \in \{1, 4, 16, 64, 256, 1024\}$ and fetch factors $f \in \{1, 4, 16, 64, 256, 1024\}$. Each configuration was warmed up for 30 seconds followed by 120 seconds of measurement. AnnLoader, the standard PyTorch data loader for AnnData, served as the baseline.

Figure 2 shows that scDataset's throughput increases substantially with both parameters. At $b = 1$ and $f = 1$, scDataset matches AnnLoader's baseline of ~20 samples/sec (pure random sampling). Increasing block size enables contiguous reads that modern storage systems handle efficiently, while higher fetch factors allow the HDF5 backend to batch multiple read requests. At the largest tested values ($b = 1024$, $f = 1024$), scDataset achieves over **204×** higher throughput than the baseline. Notably, throughput plateaus once the block size exceeds $m \times f$ (minibatch size $\times$ fetch factor), since at this point the entire fetch consists of a single contiguous read: larger blocks provide no additional I/O benefit when the fetch already retrieves cells sequentially.

We also benchmarked scDataset on HuggingFace Datasets and BioNeMo-SCDL formats, achieving over **47×** and **25×**

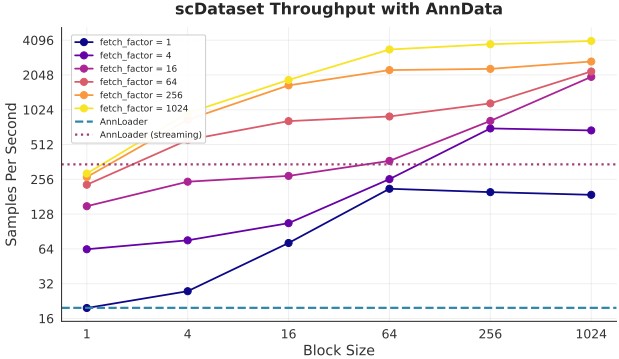

*Figure 2.* Data loading throughput on AnnData as a function of block size and fetch factor. Throughput (samples/sec) increases with both parameters, reaching **204×** speedup over AnnLoader at the largest values.

speedups respectively. These results are reported in Appendix D. Multiprocessing performance with AnnData is evaluated in Appendix E.

### 4.2. Streaming Performance and Fetch Factor

Interestingly, scDataset with block shuffling can *exceed* the throughput of pure sequential streaming from AnnData (horizontal dotted line in Figure 2). Figure 3 isolates this effect: even when reading cells sequentially without shuffling, increasing the fetch factor from 1 to 1024 yields over **15×** higher throughput than AnnLoader's streaming baseline.

This improvement follows from the fixed per-call overhead of the HDF5 backend (Section 3.2): batched fetching issues a single request for $m \times f$ cells rather than $f$ separate minibatch requests, amortizing that cost across the entire fetch. Consequently, even for inference workloads where minibatch diversity is irrelevant, a high fetch factor substantially accelerates throughput.

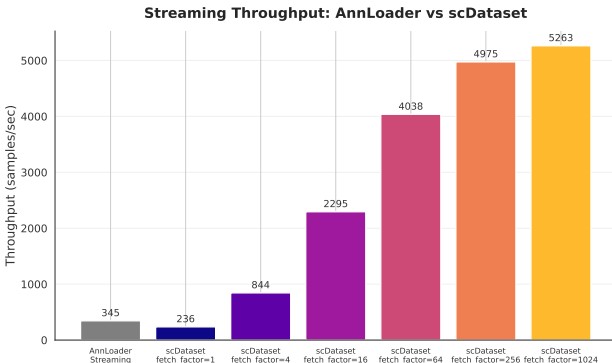

*Figure 3.* Effect of fetch factor on streaming throughput from Ann-Data. Batched fetching amortizes fixed I/O overhead, achieving over 15× speedup at $f = 1024$ compared to iterative minibatch fetching.

### 4.3. Minibatch Diversity

Block sampling introduces correlations within minibatches: cells from the same block are more likely to share metadata such as experimental plate, cell line, or drug treatment. We quantified this effect using plate label entropy within minibatches. In Tahoe-100M, each AnnData file corresponds to a unique plate label, and because files are concatenated without prior shuffling, adjacent cells share the same label. True random sampling across all 14 plates yields an empirical minibatch entropy of ∼3.62 (minibatch size 64).

Figure 4 shows how entropy varies with block size and fetch factor. As expected, increasing block size reduces entropy: at $b \geq 64$ with $f = 1$, all cells in a minibatch come from the same block, yielding entropy near zero. More generally, entropy collapses to zero whenever $b \geq m \times f$, since the entire fetch then consists of a single contiguous block from one plate, eliminating any diversity. However, batched fetching counteracts this effect by drawing cells from multiple blocks before reshuffling. With $b = 16$ and $f = 256$, entropy reaches ∼3.61, nearly matching true random sampling, because each minibatch draws from up to $(m \times f)/b = (64 \times 256)/16 = 1024$ distinct blocks after reshuffling.

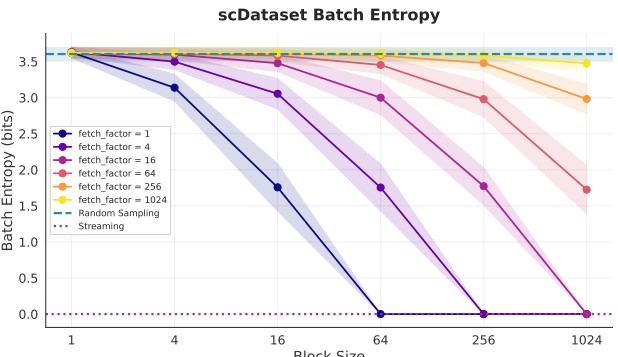

*Figure 4.* Plate label entropy within minibatches as a function of block size and fetch factor. Higher fetch factors compensate for the diversity loss from larger block sizes.

### 4.4. Real-World Classification Tasks

To validate that quasi-random sampling preserves model performance across architectures, we trained two models on four prediction tasks from Tahoe-100M: a linear classifier and a two-hidden-layer MLP (hidden size 512, GELU activations). The tasks are cell line classification (50 classes), drug classification (380 classes), and mechanism of action (MoA) classification at broad (4 classes) and fine (27 classes) resolution; MoA labels were provided by the dataset authors.

We compared four data loading strategies: (1) *Streaming* without shuffling, (2) *Streaming* with a *shuffle buffer* of 16,384 cells (64 × 256), (3) *BlockShuffling* with $b = 16$

and $f = 256$, and (4) *Random Sampling* (BlockShuffling with $b = 1$). All models were trained for one epoch using Adam (Kingma & Ba, 2015) with learning rate $1 \times 10^{-5}$. Plates 1–13 ($\sim$94 million cells) served as training; plate 14 ($\sim$6.5 million cells) served as test, containing at least one occurrence of every cell line and drug. Each experiment was repeated three times with different random seeds.

Figure 5 shows macro F1-scores (mean $\pm$ standard deviation over 3 seeds) for both architectures. Both show the same pattern across all tasks. Streaming and streaming with a shuffle buffer achieve similarly poor performance, confirming that a buffer of 16,384 cells does not mitigate bias when plate-scale heterogeneity spans tens of millions of cells. BlockShuffling with $b = 16$, $f = 256$ matches Random Sampling on all four tasks for both architectures, with near-zero variance across seeds in all configurations.

The magnitude of the benefit depends on the degree to which plate-level structure confounds the learning signal. Cell line identity is preserved across all experimental plates and provides a strong training signal regardless of sampling order; consequently, all strategies achieve comparable cell line classification performance (linear F1: 0.935–0.937; MLP F1: 0.942 across strategies). Drug and MoA classification, by contrast, require observing each cell line under diverse perturbations: when training proceeds plate by plate, the model only ever sees each cell line in the context of the drugs on that plate, systematically corrupting the learning signal. For these tasks, streaming collapses to near-chance performance while BlockShuffling matches Random Sampling.

Figure 6 shows training loss curves per task for both architectures. Streaming exhibits periodic loss spikes at plate boundaries, consistent with the catastrophic forgetting dynamics discussed in Section 1: each plate transition exposes the model to an abrupt distribution shift, driving a sharp increase in loss before the model partially adapts to the new plate. BlockShuffling and Random Sampling produce smooth loss curves throughout training, reflecting stable gradient updates from diverse minibatches.

## 5. Discussion

This work introduces **scDataset**, a scalable and flexible data loader for training deep learning models on large-scale single-cell omics datasets. By combining block sampling and batched fetching, scDataset enables randomized, high-throughput training directly from disk without requiring format conversion or full in-memory loading. The implementation integrates directly with PyTorch, provides native support for multiprocessing and distributed training, and consistently delivers substantial speed-ups over existing solutions such as AnnLoader, HuggingFace Datasets, and

BioNeMo. By operating directly on formats like AnnData, scDataset enables shuffled training on commodity hardware and lowers the barrier to large-scale deep learning in single-cell biology.

**Training dynamics and downstream implications.** The training loss curves in Section 4.4 show that the difference between streaming and quasi-random sampling is not confined to final evaluation metrics. Streaming produces sharp loss spikes at plate transitions throughout training, reflecting repeated episodes of catastrophic forgetting as the model is abruptly exposed to new experimental conditions. BlockShuffling eliminates these spikes entirely, producing training dynamics that are indistinguishable from those of true Random Sampling. For applications that involve multi-epoch training or fine-tuning of large foundation models, where ordering effects can compound across passes over the data, stable training dynamics may be as important as the final validation score.

**Generalizability beyond single-cell genomics.** While developed for single-cell omics, scDataset's approach applies broadly to any large-scale dataset where natural clustering (spatial, temporal, or organizational) and memory constraints both arise. The method extends naturally to proteomics with plate effects, spatial transcriptomics organized by tissue sections, time-series sensor data from IoT deployments, geospatial imagery organized by geographic tiles, medical imaging archives grouped by patient cohorts, among others. scDataset's modular backend interface supports arbitrary indexable collections through customizable callbacks, enabling deployment across these domains without modification.

**Future storage formats and deployment scenarios.** scDataset provides experimental support for automated profiling to recommend $(b, f)$ parameters based on dataset and hardware characteristics, though further development is needed to handle diverse organizational patterns robustly. Looking forward, scDataset's design positions it to benefit from emerging storage formats. AnnBatch, a collaborative effort between scverse and Lamin Labs, is integrating shuffled zarr-backed storage into the AnnData ecosystem. Zarr v3 offers cloud-native chunked storage with sharding, concurrent I/O, and rust-accelerated access. Preliminary benchmarks show zarr can outperform HDF5 for sequential access. The combination of scDataset's quasi-random sampling with Zarr backends could deliver best-in-class throughput for atlas-scale datasets hosted in centralized repositories like CZ CELLxGENE Census, while maintaining full ecosystem compatibility.

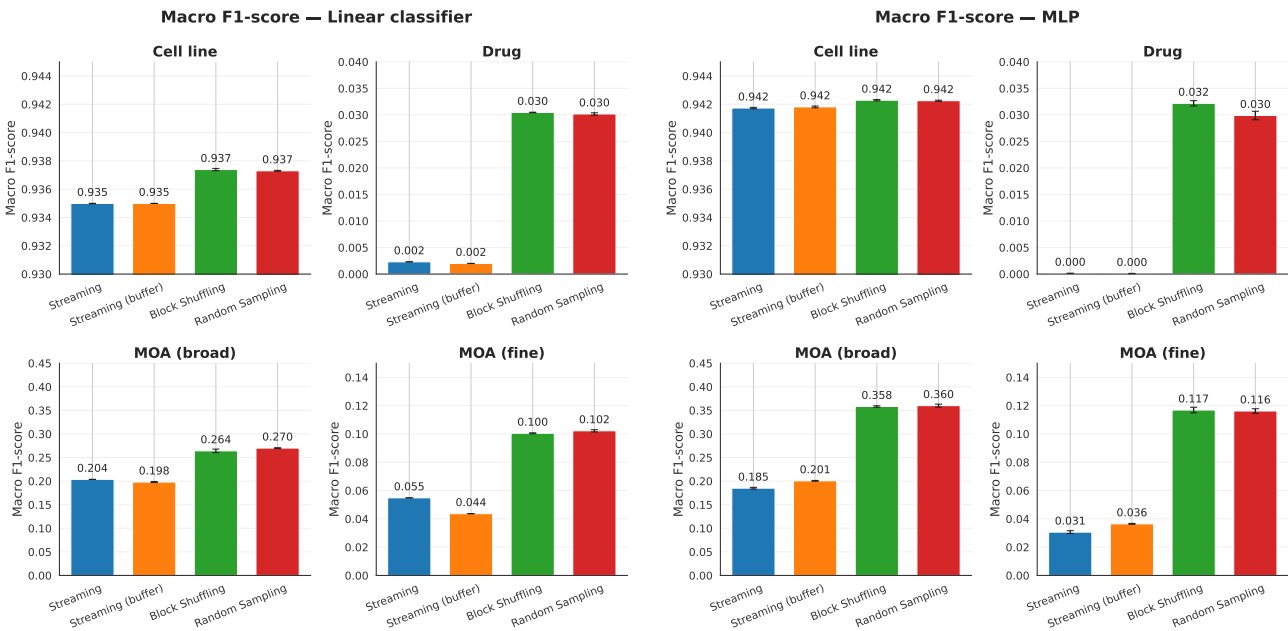

*Figure 5.* Macro F1-score (mean ± std over 3 seeds) across four tasks. *Left*: linear classifier. *Right*: two-hidden-layer MLP. BlockShuffling with $b = 16$, $f = 256$ matches Random Sampling on all tasks for both architectures; streaming strategies underperform on drug and MoA classification due to plate-scale heterogeneity.

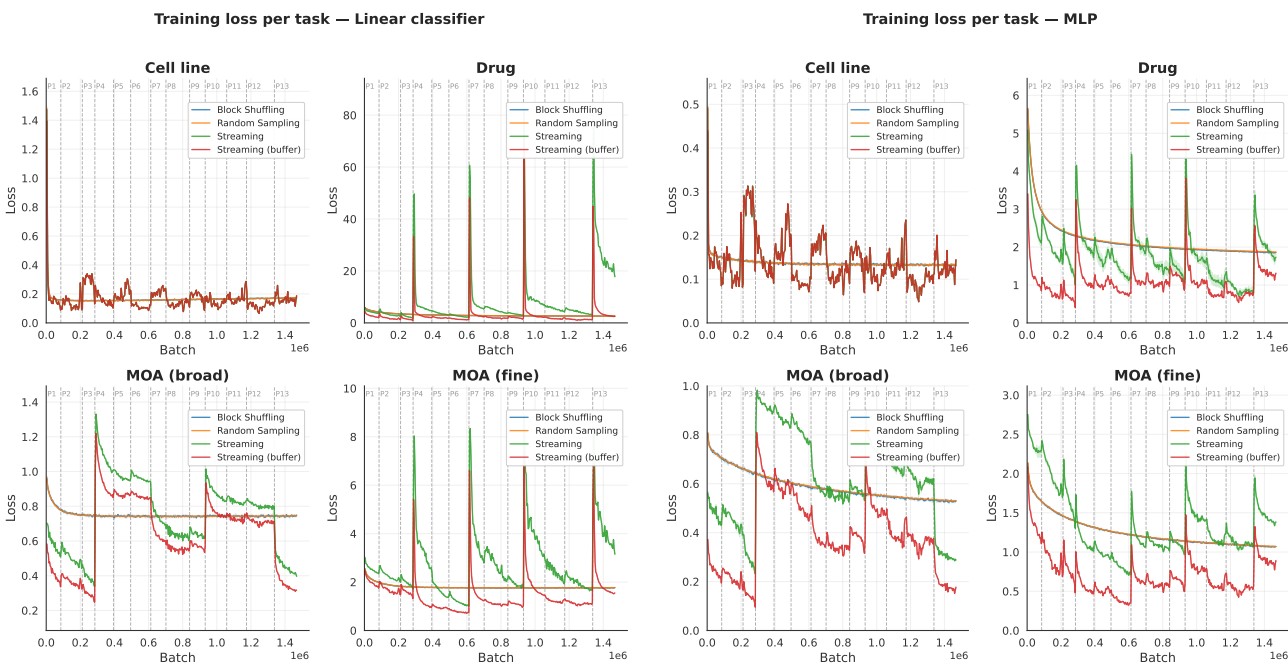

*Figure 6.* Training loss curves per task. *Left*: linear classifier. *Right*: MLP. Streaming exhibits sharp loss spikes at plate boundaries, consistent with catastrophic forgetting when the data stream transitions between experimental plates. BlockShuffling and Random Sampling produce smooth monotonic loss descent throughout training.

## Acknowledgements

Davide D'Ascenzo was financially supported by the Italian National PhD Program in Artificial Intelligence (DM 351 intervento M4C1 - Inv. 4.1 - Ricerca PNRR), funded by NextGenerationEU (EU-NGEU). Sebastiano Cultrera di Montesano was supported by the Eric and Wendy Schmidt Center at the Broad Institute of MIT and Harvard.

## Impact Statement

This work helps democratize large-scale model training to the broader scientific community.

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

## A. Data Flow and Callback Architecture

scDataset provides a flexible data transformation pipeline through four optional callback hooks. These callbacks separate data access logic from sampling logic, enabling seamless integration with arbitrary data backends without modifying the core sampling algorithms.

The data loading pipeline executes the following steps: (1) the sampling strategy generates an ordered index sequence for the epoch; (2) indices are partitioned into fetch batches of size $m \times f$; (3) `fetch_callback` retrieves raw data from the collection (default: `collection[indices]`); (4) `fetch_transform` processes the fetched chunk, e.g., sparse-to-dense conversion or materialization of backed arrays (applied once per fetch); (5) fetched data is reshuffled in memory; (6) `batch_callback` extracts a single minibatch (default: `transformed_data[batch_indices]`); (7) `batch_transform` applies final preprocessing such as normalization or tensor conversion (applied once per batch); (8) the processed batch is yielded to the training loop.

**fetch_callback**  Signature: `(data_collection, indices) -> fetched_data`. Override when the collection does not support standard indexing (e.g., database queries).

**fetch_transform**  Signature: `(fetched_data) -> transformed_data`. Efficient for chunk-level operations: materializing backed AnnData, sparse-to-dense conversion, or creating `MultiIndexable` objects for multi-modal data.

**batch_callback**  Signature:  `(transformed_data, batch_indices) -> batch`.  Rarely  needed  if `fetch_transform` produces indexable output.

**batch_transform**  Signature: `(batch) -> transformed_batch`. Handles normalization, data augmentation, or PyTorch tensor conversion.

This design separates *what* to sample from *how* to access data, enabling identical sampling algorithms to operate on AnnData, HuggingFace Datasets, TileDB-SOMA, or custom backends. It also distinguishes chunk-level operations (applied once to $m \times f$ samples) from batch-level operations (applied $f$ times to $m$ samples), allowing expensive transformations to be placed at the appropriate granularity.

### A.1. Multi-Modal Data

scDataset includes a `MultiIndexable` container for grouping multiple indexable objects (e.g., gene expression matrices, protein measurements, and cell metadata) that should be indexed together. When any of the four callbacks returns a `MultiIndexable`, subsequent indexing operations automatically synchronize across all contained arrays. This is particularly useful for multi-modal single-cell data (e.g., CITE-seq with RNA and protein modalities) where different data types must remain aligned through the batching process.

## B. Distributed Training and Weighted Sampling

Standard PyTorch workflows rely on `DistributedSampler` for distributed training and `WeightedRandomSampler` for handling class imbalance. However, these components cannot be used together: `DistributedSampler` expects to control index generation to partition data across ranks, while `WeightedRandomSampler` requires full control over sampling probabilities. This mutual exclusivity has been a known limitation since 2019[7], forcing users to choose between distributed training and principled strategies for imbalanced datasets.

scDataset resolves this conflict by separating *what* to sample from *how* to distribute work. All sampling strategies (including weighted and class-balanced variants) generate the same deterministic global index sequence on all ranks. This sequence defines the complete ordering of blocks to sample during an epoch. Work distribution then occurs at the fetch level: each rank processes fetches at positions $\text{rank}, \text{rank} + \text{world\_size}, \text{rank} + 2 \cdot \text{world\_size}, \ldots$ in round-robin fashion. For example, with 4 ranks and 100 fetches per epoch, rank 0 processes fetches $\{0, 4, 8, \ldots, 96\}$ while rank 1 processes $\{1, 5, 9, \ldots, 97\}$. This ensures that any sampling strategy works with distributed training without modification.

When DataLoader workers are enabled (`num_workers > 0`), each DDP rank spawns its own worker pool, creating a

---

[7]https://github.com/pytorch/pytorch/issues/23430

two-level parallelism hierarchy. With $R$ ranks and $W$ workers per rank, the dataset is effectively partitioned among $R \times W$ parallel processes. Workers within each rank further subdivide their rank's assigned fetches, with each worker processing approximately $1/W$ of the rank's total fetches. scDataset coordinates both levels automatically without requiring manual configuration.

To ensure reproducibility, scDataset auto-detects the distributed environment and broadcasts a shared random seed from rank 0 to all other ranks. This seed controls both the block-level permutation and the in-memory reshuffling within fetches, guaranteeing that all ranks observe the same global sampling order even when processing different data subsets.

## C. Proofs for Section 3.4

We provide proofs of Theorems C.1 and C.2 and Proposition 3.1. Throughout, we use the classical bias expansion of the plug-in entropy estimator for multinomial samples (Paninski, 2003).

### C.1. Proof of Theorem C.1

**Theorem C.1** (Large fetch factor). *As $f \to \infty$ with fixed $m, b, K$, the expected entropy satisfies*

$$\mathbb{E}[H(C)] = H(p) - \frac{K-1}{2m \ln 2} + O(m^{-2}), \tag{4}$$

*where $H(p) = -\sum_{k=1}^{K} p_k \log_2 p_k$.*

*Proof.* When $f \to \infty$, the buffer contains $fm \to \infty$ cells. By the law of large numbers, the fraction of cells from label $k$ in the buffer converges almost surely to $p_k$.

Selecting $m$ cells uniformly from this large buffer is asymptotically equivalent to drawing $m$ IID samples from $\mathrm{Cat}(p)$. Therefore,

$$(C_1, \ldots, C_K) \xrightarrow{d} \mathrm{Multinomial}(m, p).$$

For multinomial samples, the expected plug-in entropy admits the expansion (Paninski, 2003)

$$\mathbb{E}[H(C)] = H(p) - \frac{K-1}{2m \ln 2} + O(m^{-2}),$$

which yields the stated result. $\square$

### C.2. Proof of Theorem C.2

**Theorem C.2** (No batched fetching). *When $f = 1$,*

$$\mathbb{E}[H(C)] = H(p) - \frac{K-1}{2B \ln 2} + O(B^{-2}), \qquad B = \frac{m}{b}. \tag{5}$$

*Proof.* When $f = 1$, exactly $B = m/b$ blocks are sampled. Let $N_k$ denote the number of blocks drawn from label $k$. Then

$$(N_1, \ldots, N_K) \sim \mathrm{Multinomial}(B, p),$$

and each block contributes $b$ cells from its label, so that

$$C_k = b\,N_k.$$

The empirical frequencies satisfy

$$\frac{C_k}{m} = \frac{bN_k}{bB} = \frac{N_k}{B}.$$

Therefore the minibatch entropy is exactly the plug-in entropy of $B$ multinomial samples. Applying the standard bias expansion with effective sample size $B$ gives

$$\mathbb{E}[H(C)] = H(p) - \frac{K-1}{2B \ln 2} + O(B^{-2}),$$

which proves the claim. $\square$

## C.3. Proof of Proposition 3.1

*Proof.* For any $f \geq 1$, the sampling procedure interpolates between the two extreme regimes.

When $f = 1$, Theorem C.2 gives

$$\mathbb{E}[H(C)] = H(p) - \frac{K-1}{2B \ln 2} + O(B^{-2}) = H(p) - \frac{(K-1)b}{2m \ln 2} + O(m^{-2}).$$

When $f \to \infty$, Theorem C.1 gives

$$\mathbb{E}[H(C)] = H(p) - \frac{K-1}{2m \ln 2} + O(m^{-2}).$$

For intermediate $f$, increasing $f$ increases the effective number of independent blocks contributing to each minibatch, and therefore monotonically increases the effective sample size from $B$ toward $m$. This yields the sandwich bound

$$H(p) - \frac{(K-1)b}{2m \ln 2} \;\leq\; \mathbb{E}[H(C)] \;\leq\; H(p) - \frac{K-1}{2m \ln 2},$$

as claimed. □

# D. Alternative Storage Backends

While the main text focuses on AnnData, we also evaluated scDataset on two alternative storage formats commonly used for large-scale single-cell data: HuggingFace Datasets and BioNeMo-SCDL. These experiments demonstrate scDataset's backend-agnostic design and reveal how storage format affects the benefits of block sampling and batched fetching.

## D.1. Dataset Preparation

The Tahoe-100M dataset is available in multiple formats with varying storage requirements:

- **AnnData**: 314GB (14 files, native format)

- **HuggingFace Datasets**: 1.9TB (Parquet format)[8]

- **BioNeMo-SCDL**: 1.1TB (memory-mapped NumPy arrays, 2.2TB peak during conversion)[9]

For HuggingFace Datasets, a custom script was used to load the expression matrix. BioNeMo-SCDL stores only the expression matrix, requiring custom logic for metadata handling.

## D.2. Throughput Results

Figure 7 and Figure 8 show throughput as a function of block size and fetch factor for HuggingFace Datasets and BioNeMo-SCDL, respectively.

Both backends show a different pattern than AnnData: throughput increases only with block size, while fetch factor has no effect. This behavior stems from the absence of a batched indexing interface in these backends. Instead, each index access is served independently, so the benefits of batched fetching (coalesced I/O) are not realized. Moreover, increasing fetch factor slightly degrades throughput due to the computational overhead of in-memory shuffling and buffer management for large objects. Nonetheless, block sampling alone yields substantial speedups: **47×** for HuggingFace Datasets and **25×** for BioNeMo-SCDL at the largest block sizes.

# E. Multiprocessing Throughput Evaluation

We evaluated the throughput of scDataset on the Tahoe-100M AnnData dataset using multiprocessing, a feature not natively supported by AnnLoader. The hyperparameter search space is summarized in Table 1, with full results in Table 2. To isolate

---

[8]https://huggingface.co/datasets/tahoebio/Tahoe-100M

[9]Generated using the official conversion script: https://docs.nvidia.com/bionemo-framework/2.6/API_reference/bionemo/scdl/scripts/convert_h5ad_to_scdl/

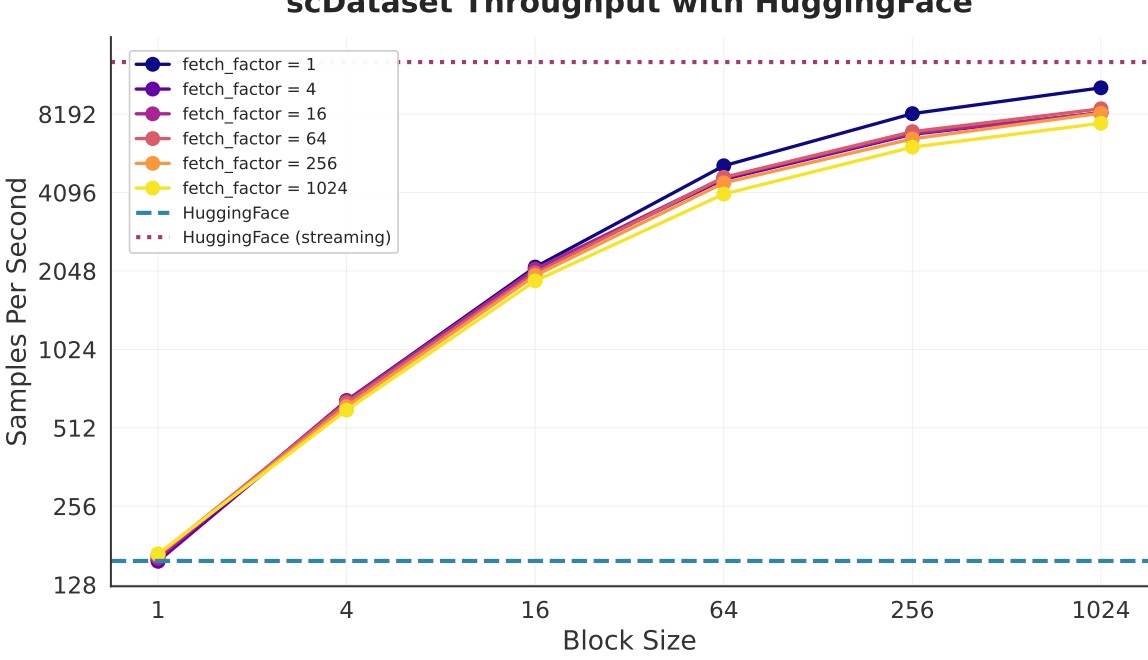

*Figure 7.* Data loading throughput on HuggingFace Datasets. Throughput scales only with block size; fetch factor has no effect due to the backend's index access pattern.

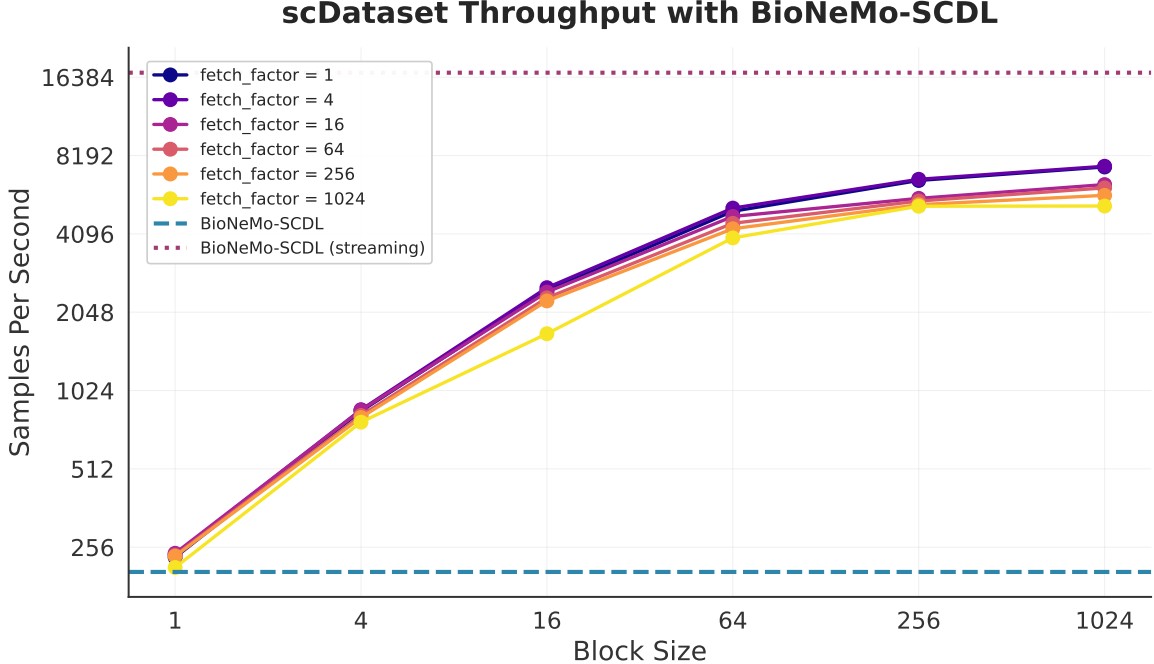

*Figure 8.* Data loading throughput on BioNeMo-SCDL. Similar to HuggingFace, throughput scales with block size while fetch factor provides no additional benefit.

the efficiency gains from parallelization beyond simple memory scaling, we compare two configurations with equivalent total buffer memory consumption.

Consider the multiprocessing configuration with `block_size=16`, `fetch_factor=256`, and `num_workers=4` (highlighted in bold in Table 2). Each worker maintains a buffer of $64 \times 256 = 16{,}384$ cells, yielding a total buffer of $16{,}384 \times 4 = 65{,}536$ cells across all workers. This configuration achieves 4614 samples/sec. For comparison, the single-core configuration with `block_size=16` and `fetch_factor=1024` in Figure 2 uses an identical buffer of $64 \times 1024 = 65{,}536$ cells but achieves only 1854 samples/sec.

Despite identical memory consumption, the multiprocessing configuration delivers a **2.5×** speedup. This improvement arises from two sources: (1) data transformations such as sparse-to-dense conversion are parallelized across CPU cores, reducing preprocessing overhead, and (2) multiple workers issue concurrent I/O requests to the HDF5 backend, which can be reordered and coalesced by the operating system to optimize disk access patterns. In contrast, a single worker must process data transformations and I/O requests serially.

*Table 1.* Hyperparameter search space for throughput experiments with multiprocessing on the Tahoe-100M AnnData dataset.

| Parameter | Values |
|---|---|
| Block size ($b$) | 4, 16, 64, 256 |
| Fetch factor ($f$) | 4, 16, 64, 256 |
| Number of workers | 4, 8, 12, 16 |

*Table 2.* Results for throughput experiments with multiprocessing on the Tahoe-100M AnnData dataset. The experiment highlighted in **bold** corresponds to the configuration used for the comparison.

| Block size | Fetch factor | Num workers | Samples/sec | Avg. batch entropy | Std. batch entropy |
|---|---|---|---|---|---|
| 4 | 4 | 4 | 289 | 3.51 | 0.11 |
| | | 8 | 575 | 3.51 | 0.11 |
| | | 12 | 866 | 3.50 | 0.11 |
| | | 16 | 1153 | 3.50 | 0.11 |
| | 16 | 4 | 918 | 3.59 | 0.08 |
| | | 8 | 1824 | 3.59 | 0.08 |
| | | 12 | 2652 | 3.59 | 0.08 |
| | | 16 | 3531 | 3.59 | 0.08 |
| | 64 | 4 | 2080 | 3.61 | 0.08 |
| | | 8 | 3850 | 3.61 | 0.08 |
| | | 12 | 4273 | 3.62 | 0.07 |
| | | 16 | 4299 | 3.62 | 0.07 |
| | 256 | 4 | 2981 | 3.62 | 0.07 |
| | | 8 | 4338 | 3.62 | 0.07 |
| | | 12 | 4401 | 3.62 | 0.07 |
| | | 16 | 4180 | 3.62 | 0.07 |
| **16** | 4 | 4 | 417 | 3.03 | 0.22 |
| | | 8 | 839 | 3.03 | 0.22 |
| | | 12 | 1195 | 3.03 | 0.22 |
| | | 16 | 1603 | 3.03 | 0.22 |
| | 16 | 4 | 1118 | 3.46 | 0.13 |
| | | 8 | 2200 | 3.48 | 0.12 |
| | | 12 | 3270 | 3.48 | 0.12 |
| | | 16 | 3851 | 3.48 | 0.12 |
| | 64 | 4 | 3156 | 3.58 | 0.09 |
| | | 8 | 4349 | 3.58 | 0.09 |
| | | 12 | 4453 | 3.59 | 0.08 |
| | | 16 | 4399 | 3.59 | 0.08 |
| | **256** | **4** | **4614** | **3.61** | **0.07** |
| | | 8 | 4562 | 3.61 | 0.08 |
| | | 12 | 4403 | 3.62 | 0.07 |
| | | 16 | 4470 | 3.61 | 0.08 |
| 64 | 4 | 4 | 928 | 1.81 | 0.30 |
| | | 8 | 1788 | 1.79 | 0.30 |
| | | 12 | 2629 | 1.87 | 0.30 |
| | | 16 | 3576 | 1.79 | 0.30 |
| | 16 | 4 | 1577 | 3.01 | 0.21 |
| | | 8 | 3110 | 2.99 | 0.22 |
| | | 12 | 4163 | 3.00 | 0.23 |
| | | 16 | 4141 | 2.97 | 0.23 |
| | 64 | 4 | 3737 | 3.47 | 0.12 |
| | | 8 | 4429 | 3.46 | 0.13 |
| | | 12 | 4494 | 3.45 | 0.12 |
| | | 16 | 4478 | 3.47 | 0.12 |
| | 256 | 4 | 4615 | 3.58 | 0.08 |
| | | 8 | 4583 | 3.58 | 0.09 |
| | | 12 | 4601 | 3.59 | 0.09 |
| | | 16 | 4604 | 3.58 | 0.09 |
| 256 | 4 | 4 | 1846 | 0.41 | 0.45 |
| | | 8 | 3691 | 0.11 | 0.29 |
| | | 12 | 4100 | 0.21 | 0.39 |
| | | 16 | 4035 | 0.42 | 0.46 |
| | 16 | 4 | 3126 | 1.84 | 0.31 |
| | | 8 | 4339 | 1.89 | 0.29 |
| | | 12 | 4412 | 1.73 | 0.34 |
| | | 16 | 4417 | 1.87 | 0.29 |
| | 64 | 4 | 4456 | 2.99 | 0.21 |
| | | 8 | 4501 | 2.99 | 0.22 |
| | | 12 | 4589 | 2.98 | 0.22 |
| | | 16 | 4569 | 2.99 | 0.22 |
| | 256 | 4 | 4575 | 3.44 | 0.13 |
| | | 8 | 4634 | 3.47 | 0.12 |
| | | 12 | 4600 | 3.46 | 0.12 |
| | | 16 | 4631 | 3.47 | 0.13 |

