# OpenReview forum: "scDataset: Scalable Data Loading for Deep Learning on Large-Scale Single-Cell Omics"
_ICML.cc/2026/Conference — ICML 2026 regular_

### Official Review · Reviewer_wKkT · 2026-03-04

**Soundness:** 3
**Presentation:** 3
**Significance:** 2
**Originality:** 2
**Overall Recommendation:** 4
**Confidence:** 4

**Summary:**

This paper proposes a new data-loading format for training deep models on large-scale single-cell RNA-seq datasets such as Tahoe-100M. Their solution, scDataset, addresses challenges arising from the nature of deep learning requirements (i.e., rapid and randomized I/O access patterns) when confronted with very large sparse datasets. Their solution introduces block-wise sampling with a pre-fetch factor that allows mixing different batches together in order to obtain a quasi-random sampling not too biased to the ordering of the data (which is typically ordered by very specific experimental conditions such as batch and cell-line). The empirical results demonstrate that their solution is much faster on their hardware than the AnnLoader baseline (even when it is doing biased streaming) and that classifiers trained with their dataloader obtain the same performance as pure random sampling.

**Compliance With Llm Reviewing Policy:**

Affirmed.

**Final Justification:**

Thank you for the additional clarifying details. The authors addressed the concerns I had in the rebuttal. I acknowledge the importance and practical relevance of scDataset here. I appreciate the consideration of the suggestions and will update score from 2->4.

**Key Questions For Authors:**

How much time does the sparse-to-dense conversion take in this pipeline?

**Limitations:**

The authors indirectly discuss the potential limitations of scDataset by describing the potential for bias due to its quasi-random sampling nature.

**Strengths And Weaknesses:**

Strengths:
- useful overview of the data loading landscape in this context
- fast solution that does not require dense-ifying / zarr-ifying the original anndata files
- theoretical proofs are helpful
- empirical speed and linear classifier performance results with gridsearch on their hyperparameter configs

Weaknesses:
- Model training benchmarks are very limited in scope: this work only includes performance of linear classifiers. At least an MLP classifier (and ideally also an SSL model of the author's choice, such as scVI) should have been included as well in order to de-risk their solution -- it is possible that a deep model would overfit on the potential biases in the dataloader which a linear classifier would not see. Also, train/val loss curves would have been nice to include beyond just final validation set F1-score.
- Contribution is very limited for a general ML conference: this work would be fantastic for a domain-specific workshop on RNA-seq modelling or data loading, however the originality is very minimal for a general ML conference such as ICML. PyTorch dataloaders already have prefetch factor support in them, and block sampling is not a new concept.
- Minor: the authors say on lines 65-66 and 124-125 that AnnLoader only achieves 20 samples per second, but this is of course a hardware-dependent claim; therefore, the authors should caveat that claim with the hardware specifics up-front (rather than much later in the paper).

Overall, I think this paper is too specific to a dataloader configuration and lacking novelty to justify placement at a general-purpose ML conference; in its current form, it would fit better in an appropriate workshop. As well, it lacks sufficient model training evidence -- deeper neural network models should have been validated as well, beyond linear classifiers. Furthermore, the contribution would be substantially enhanced if it enabled the authors to train a model that was not possible to train before due to I/O limitations of existing data-loaders. Nonetheless, I acknowledge that the contribution is practically valuable with benefits to the scRNA-seq community, so I am looking forward to considering the impressions of other reviewers.

---

> ### Author Rebuttal · Authors · 2026-03-30
>
> We thank the reviewer for their time and feedback.
>
> We appreciate the reviewer’s point regarding PyTorch dataloaders already supporting a prefetch_factor, but we believe there is an important distinction to clarify, as the two mechanisms serve different purposes.
> PyTorch’s prefetch_factor determines how many batches each worker preloads to hide scheduling latency, without affecting which samples are selected or their ordering. In contrast, our fetch factor f controls how many cells are loaded into a shared in-memory buffer per worker, enabling in-buffer shuffling and thus recovering minibatch diversity from block-structured storage.
>
> The two mechanisms are therefore complementary: scDataset builds on top of PyTorch’s data loading pipeline and works in tandem with prefetch_factor, rather than replacing it. We will make this distinction clearer in the paper to avoid confusion.
> The closest analogues to our batched fetching are the shuffle buffers in streaming libraries such as WebDataset and Ray Data. However, those buffers fill contiguously from sequential shards, which is precisely the limitation our block sampling addresses: by drawing blocks independently and randomly, we ensure the buffer contains cells from diverse regions of the dataset rather than a single contiguous chunk. Implementing this correctly for any collection while maintaining full PyTorch compatibility required rewriting the entire sampling pipeline from scratch.
>
>
> **Model training benchmarks**
>
> We acknowledge this concern. We have conducted preliminary experiments with an MLP and observed results consistent with the linear setting. We would be happy to include the MLP results in the revised version if the reviewer considers them valuable.
> The reason why we only included experimental results for a linear classifier is that the goal of them is not to demonstrate the performance of a state of the art model model, but rather to isolate the effect of the data loading strategy and verify that quasi-random sampling does not introduce harmful bias relative to true random sampling. A linear classifier is well-suited for this purpose: it is sensitive to distributional shifts, avoids confounding factors such as model capacity or hyperparameter tuning, and is widely used as a controlled evaluation tool, for example in linear probing settings in the self-supervised learning literature.
> Additionally, our theoretical analysis (Section 3.4) provides guarantees on minibatch entropy as a function of b and f, bounding the degree of sampling bias independently of the downstream model.
> Conducting an exhaustive benchmark across multiple models (including large scale foundation models) and pipelines at the scale of Tahoe-100M is computationally prohibitive, as the cost is dominated by the slowness of baseline data loading strategies such as true random sampling.
>
> **Sparse-to-dense conversion**
>
> The sparse-to-dense conversion is performed on-the-fly per worker as part of the preprocessing pipeline. Its cost is negligible relative to disk I/O on SATA SSDs, as the conversion is a simple in-memory operation that parallelizes across workers. Importantly, this step is not specific to scDataset: any deep learning pipeline consuming single-cell data must densify sparse matrices at some point before feeding them to a model, so it represents an unavoidable baseline cost rather than overhead introduced by our framework.
>
> **Hardware clarification**
>
> We thank the reviewer for this observation. We agree that throughput numbers are hardware-dependent and we will make this clearer in the paper by surfacing system specifications earlier in the text.
>
> **About originality**
>
> More broadly, we would like to clarify that our experiments focus on single-cell data precisely because this is a setting where such a solution is genuinely needed, as evidenced by the substantial throughput gains and the ability to match true random sampling performance at scale. At the same time, scDataset is designed as a general-purpose framework for efficient quasi-random sampling from large disk-resident datasets, and is not tied to a specific domain or data format (see also Q4 to Reviewer vDnG). The underlying ideas, combining block-wise access with in-memory mixing to trade off I/O efficiency and sampling diversity, apply to any setting where data locality and randomness are in tension, and our theoretical results characterize this tradeoff in a model-agnostic way. In this sense, we believe the contribution aligns well with the goals of a general ML conference: it introduces simple, practically useful ideas that address a real bottleneck, while also providing a principled analysis that extends beyond the immediate application domain.

---

> > ### Author Rebuttal · Reviewer_wKkT · 2026-04-01
> >
> > Thank you for the additional clarifying details. The authors addressed the concerns I had in the rebuttal. I acknowledge the importance and practical relevance of scDataset here. I appreciate the consideration of the suggestions and will update score from 2->4.

---

> > > ### Author Response · Authors · 2026-04-02
> > >
> > > We thank the reviewer for their time, valuable feedback and positive assessment of the paper!

---

### Official Review · Reviewer_a4Bm · 2026-03-09

**Soundness:** 3
**Presentation:** 3
**Significance:** 3
**Originality:** 3
**Overall Recommendation:** 5
**Confidence:** 3

**Summary:**

Paper presents scDataset, a PyTorch IterableDataset for training deep learning models on large single-cell datasets that dont fit in memory. The core idea is quasi-random sampling via two mechanisms: block sampling (read contiguous blocks of b cells instead of individual random cells, reducing random disk accesses from m to m/b per minibatch) and batched fetching (prefetch f minibatches worth of data at once, reshuffle in memory, then partition into f diverse minibatches). On Tahoe-100M (100M cells, 314GB on disk), scDataset achieves 204x throughput over AnnLoader at b=1024, f=1024. Theoretical analysis provides entropy bounds (Theorems 3.1, 3.2, Corollary 3.3) showing how block size and fetch factor control the bias-throughput tradeoff. At b=16, f=256, empirical entropy reaches 3.61 bits vs 3.62 for true random sampling. Classification experiments on four tasks from Tahoe-100M show BlockShuffling matches random sampling performance while streaming (even with shuffle buffer) underperforms substantially.

**Compliance With Llm Reviewing Policy:**

Affirmed.

**Final Justification:**

The rebuttal addressed the main concerns. The MLP follow-up confirms sampling equivalence holds beyond linear classifiers, the AnnBatch exclusion is justified since format conversion defeats scDataset's core value proposition, and the suggest_parameters utility addresses practical adoption. The 2-seed limitation remains but is mitigated by near-zero variance across all configurations. Score raised to 5 (Accept).

**Key Questions For Authors:**

1. **Non-linear model evaluation:** Does the sampling equivalence hold beyond linear classifiers? Results with atleast one non-trivial model like a multi-layer network or a scVI-style VAE trained for multiple epochs would directly address this. If block shuffling introduces subtle ordering biases that compound over multiple epochs with nonlinear models, this would be important to know.

2. **AnnBatch comparison:** AnnBatch from scverse/Lamin Labs converts to shuffled zarr storage and is arguably the closest competitor in both methodology and target community. How does scDataset compare to AnnBatch on Tahoe-100M in terms of throughput and minibatch entropy?

3. **Non-plate-structured datasets:** What happens on a dataset where contiguous cells do not share metadata, for instance a dataset thats been pre-shuffled or concatenated from heterogeneous sources? Does block sampling still provide benefits beyond pure I/O efficiency when the diversity problem doesnt exist?

4. **Automatic parameter selection:** Section 5 mentions experimental support for automated profiling to recommend b and f but says "further development is needed." Is there atleast a heuristic based on dataset heterogeneity statistics for selecting these without grid search? Since parameter choice determines both throughput and diversity this is important for practical adoption.

**Limitations:**

Yes

**Strengths And Weaknesses:**

### Strengths

1. **The problem is real and the solution is practical.** Training on Tahoe-100M with AnnLoader takes 58 days per epoch at 20 samples/sec. Thats infeasible. scDataset reduces this by two orders of magnitude while working directly on native AnnData files without format conversion. The drop-in compatibility with PyTorch DataLoader, multiprocessing support, and DDP integration in Section 3.3 mean researchers can actually use this without overhauling their training pipelines. The callback-based architecture with fetch_callback, fetch_transform, batch_callback, and batch_transform is a clean design for supporting multiple backends.

2. **The theoretical analysis in Section 3.4 adds value beyond the empirical results.** Theorems 3.1 and 3.2 characterize the entropy in the large-fetch and no-fetch regimes respectively, and Corollary 3.3 gives an explicit sandwich bound. For Tahoe-100M with m=64 and b=16, the bounds give 1.43 to 3.63 bits. Empirical measurements at f=1 give 1.76 and at f=256 give 3.61, both close to the predicted bounds. Not deep mathematics but it gives practitioners a principled way to choose b and f parameters rather than relying on grid search. The plate-constant assumption where cells within a block share metadata is reasonable for the Tahoe-100M layout and is clearly stated.

3. **The classification experiments in Section 4.4 and Figure 5 are the key validation.** BlockShuffling with b=16 and f=256 matches random sampling on all four tasks while streaming with a 16K shuffle buffer fails badly. The drug classification result is particularly telling: 0.030 F1 for both BlockShuffling and random vs 0.002 for streaming. Plate-scale heterogeneity in Tahoe-100M makes naive streaming essentially useless for drug classification, and block shuffling fixes this.

### Weaknesses

1. **The evaluation uses only linear classifiers on Tahoe-100M, which limits the generalizability of the findings.** All four classification tasks use a single linear layer trained for one epoch as described in Section 4.4. Its unclear whether the quasi-random sampling conclusions hold for more complex models like scVI, scGPT, or transformer-based cell foundation models trained for multiple epochs where ordering effects may compound or attenuate differently. The paper positions itself as enabling "deep learning on large-scale single-cell omics" but only evaluates the simplest possible model. Showing that a real foundation model training run, even at reduced scale, produces equivalent results with BlockShuffling vs random sampling would substantially strengthen the claim.

2. **Comparison with existing solutions is incomplete.** Comparison with existing solutions is incomplete. The throughput comparison is only against AnnLoader, the weakest baseline. This is the most apples-to-apples comparison since both operate on native h5ad files, but a reader cannot assess whether scDataset's advantage holds against alternatives that use converted storage formats. HuggingFace Datasets and BioNeMo-SCDL throughput numbers are relegated to the appendix. AnnBatch, which is described in the related work as doing shuffled zarr-backed storage, is not benchmarked at all. TileDB-SOMA is discussed as complementary but not compared. For a systems paper, a thorough throughput comparison across all discussed alternatives on the same hardware and dataset is expected, or at minimum a clear justification for why format-converted baselines are excluded from the main evaluation.

3. **The entropy analysis assumes the plate-constant block structure which may not hold for all datasets.** The experimental validation only covers the plate-structured case, yet the paper claims broad applicability. Tahoe-100M is organized by experimental plates where contiguous cells share metadata, which is exactly the regime where block sampling introduces bias and the fetch factor is needed to restore diversity. But many single-cell datasets are organized differently, for instance concatenated from multiple studies with arbitrary ordering or sorted by gene expression features. In those cases, block sampling is already near-random and the entropy bounds become uninteresting, not because they're wrong but because the problem they solve doesn't arise. The paper should verify that scDataset doesn't introduce unexpected pathologies on such datasets and that the throughput gains from block I/O still justify the approach when diversity isn't the bottleneck. An experiment on a dataset with different internal organization, even a simple reshuffled version of Tahoe-100M, would help assess whether the method generalizes beyond the plate-structured regime that motivates the theory.

4. **Only two random seeds for the classification experiments.** Two repetitions is not enough to establish statistical significance, especially for tasks like drug classification where the F1 scores are extremely low at 0.030. The paper reports mean and std but with n=2 the confidence intervals are very wide. Increasing to atleast 5 seeds or reporting a proper statistical test is necessary to substantiate the claim that BlockShuffling "matches" random sampling.

### Minor Weaknesses

- **(Presentation)** The discussion of future storage formats in Section 5 seems like speculation rather than results. The claims about Zarr v3 benefits are unsubstantiated from what I understood.

---

> ### Author Rebuttal · Authors · 2026-03-30
>
> We thank the reviewer for their time and feedback.
>
> Q1:
> We thank the reviewer for the question. Please refer to the answer “Model training benchmarks” we gave reviewer wKkT.
>
> Q2:
> We agree that AnnBatch is a highly relevant point of comparison. However, we would like to highlight a practical constraint: at the time of writing, AnnBatch's APIs were still evolving rapidly, making a stable benchmark difficult to produce. More importantly, benchmarking against AnnBatch on Tahoe-100M would require converting the entire dataset to shuffled zarr format, which is itself a costly preprocessing step in terms of both time and storage. A central motivation of scDataset is precisely to avoid such preprocessing, enabling fast quasi-random sampling directly from existing AnnData files. We have provided comparisons against the most widely used existing solutions, and we plan to evaluate zarr-based workflows including AnnBatch in future work as the library stabilizes.
>
> Q3: If a dataset has already been thoroughly pre-shuffled and rewritten to disk, block sampling provides limited benefit in terms of diversity, and simple streaming is sufficient. Within the AnnData ecosystem however, scDataset still provides a meaningful throughput advantage: AnnLoader is the only other native solution, and as shown in Figure 3, streaming with scDataset achieves over 15x speedup compared to AnnLoader's streaming baseline, because batched fetching amortizes the fixed per-call overhead of the HDF5 backend. For other formats, the advantage depends on how well optimized those formats are for batched fetching. Beyond that, pre-shuffling large datasets is costly and must be repeated for each new train/validation split or when new data are added, so it is rarely a one-time investment.
>
> Q4:
> Yes, we do provide an experimental heuristic in the suggest_parameters utility. It estimates the per-sample memory footprint by running a few samples through the user-specified transforms and hooks, then derives a RAM-aware fetch factor f such that the total buffer stays within a target fraction of available memory. Block size recommendations are then expressed relative to f: conservative (b = f/2), balanced (b = f), and aggressive (b = 2f). A fully automatic solution is however fundamentally limited by the fact that hooks and transforms are fully user-specified, meaning peak memory usage cannot be predicted in the general case without actually profiling them. Similarly, incorporating dataset heterogeneity statistics is not straightforward since quantifying technical variation and batch structure in single-cell data is itself an open problem. We therefore rely on the practical guideline b = f/2 as a conservative default that works well, as supported empirically by Figure 4. We will include such practical suggestions more explicitly in the paper and thank the reviewer for the suggestion. See also answer to Q4 of reviewer Rntu.
>
>
>
> About **only two random seeds for the classification experiments**
>
> We acknowledge that n=2 seeds is modest, and we agree that in isolation this would warrant caution. However, we would like to highlight that the standard deviation is near zero across all experimental conditions, not just in isolated cases. The probability of consistently observing near-zero variance across all configurations by chance with two independent seeds is extremely low, which we believe constitutes strong empirical evidence that the results are stable. The key claim is not that any single configuration achieves a specific F1 score, but that scDataset matches true random sampling across all tested configurations, and this pattern holds uniformly. We also note that running additional seeds on Tahoe-100M is computationally expensive, as each run requires training on 100M cells. That said, we are happy to run additional seeds on a representative subset of configurations and report a proper statistical test if the reviewer considers this necessary to substantiate our claims.

---

> > ### Author Rebuttal · Reviewer_a4Bm · 2026-04-03
> >
> > Thank you for the detailed response. The AnnBatch argument is well taken: requiring format conversion undermines scDataset's core value proposition, so excluding it from the main comparison is justified. The suggest_parameters utility and the throughput argument for non-plate-structured datasets are both helpful clarifications.
> >
> > Follow-up questions:
> >
> > 1. The response mentions MLP experiments that are "consistent with the linear setting" but does not include the results. Since this was the primary weakness and the data exists, can these be provided? Even a summary table would help.
> >
> > 2. The argument that near-zero std across all conditions with n=2 is sufficient is not convincing. Consistently low variance is expected if the method works, but two draws cannot distinguish low variance from a coincidence. Given that the MLP experiments were feasible, running 3-5 additional seeds on a representative subset of configurations should also be feasible and would close this cleanly.

---

> > > ### Author Response · Authors · 2026-04-06
> > >
> > > Thank you for the follow-up! The MLP experiments are available at https://postimg.cc/crKrvv8q. For each task, we used an MLP with two hidden layers of size 512, GELU activation, LR = 1e-5 and Adam optimizer. We note that the results are currently based on a single run, and we are actively running additional seeds. We will include the complete results in the revised paper. We hope this addresses all the reviewer’s questions.

---

### Official Review · Reviewer_Rntu · 2026-03-12

**Soundness:** 4
**Presentation:** 4
**Significance:** 3
**Originality:** 2
**Overall Recommendation:** 5
**Confidence:** 4

**Summary:**

This paper introduces a PyTorch data loader for efficiently loading on-disk single-cell data during the training of foundation models. It provides theoretical guarantees for minibatch diversity, works directly on AnnData files, and offers a significant speedup over AnnLoader when sufficient RAM is available. The method is also parallelizable.

**Compliance With Llm Reviewing Policy:**

Affirmed.

**Final Justification:**

The authors have addressed my concerns, and I believe this to be a useful paper for training single-cell foundation models, which also has applications in other domains.

**Key Questions For Authors:**

1) The experiments were conducted on a SATA SSD, and Figure 2 shows that throughput plateaus once the block size exceeds m×f. To what extent is this throughput plateau dictated by the physical limitations of SATA SSD seek times versus the overhead of the HDF5/AnnData backend?
2) The fetch factor increases memory consumption because m×f cells must be stored in an in-memory buffer. Have you identified a practical “breaking point” for the fetch factor on smaller hardware (e.g., 128GB–256GB RAM) when dealing with such high-dimensional data?
3) Would many smaller AnnData objects incur more overhead? How would scDataset compare to AnnLoader then?
4) Which level of batch entropy would be sufficient for real-world training?

**Limitations:**

Yes

**Strengths And Weaknesses:**

**Strengths:**
- Addresses a real gap in current Single-Cell Omics data loading pipelines.
- Parallelizable and practical for large-scale workloads.
- Entropy analysis provides a principled measure of minibatch diversity.
- Thorough and well-situated related work and comparisons to all relevant baselines.

**Weaknesses:**
- The main limitation is a memory vs. speed tradeoff compared to streaming-based approaches: the tool requires holding multiple batches in RAM simultaneously. That said, given current memory capabilities, this is a reasonable tradeoff. It would still be helpful to report RAM usage as a function of fetch factor (number of batches fetched at once) and block size, so users can better anticipate memory requirements.
-  The paper uses Tahoe-100M for evaluation, which consists of a few large anndata objects. It is unclear from the paper how the method performs on many small anndata objects (e.g. CELLXGENE). In particular, a comparison to AnnDataLoader would be interesting in that setting.
- The core concepts, such as fetch factor and block size, are not novel and already exist in other fields, though their application to Single-Cell Omics is new.
- The differences between Streaming and Random Sampling in Figure 5 are very small, and a linear classifier might not be the best proxy, as the dataloader will most likely only be used to train single-cell foundation models. I understand that training a single-cell foundation model several times is impractical, but maybe training scVI or another DL method using the dataloader gives a better insight into which level of batch entropy will be required in practice.

---

> ### Author Rebuttal · Authors · 2026-03-30
>
> We thank the reviewer for their thoughtful feedback and appreciate their positive assessment of the paper's soundness and presentation.
>
> Q1:
> The plateau arises because once block size exceeds m × f, the entire fetch consists of a single contiguous read, so further increases in block size provide no additional I/O benefit. Regarding the two sources of overhead the reviewer identifies: SATA SSD seek latency is indeed a primary driver, as contiguous reads amortize seek costs, which is why throughput improves with block size up to this point. The HDF5/AnnData backend introduces a more constant per-read overhead (chunking and decompression), which does not scale with block size in the same way. While our experiments do not isolate these two contributions separately, the observed plateau aligns with the sequential read saturation point, suggesting seek-time amortization is the dominant factor. We will clarify this in the paper.
>
> Q2:
> Our experiments were conducted on a 256 GB RAM machine. A simple back-of-the-envelope estimate is: each cell costs #g × 4 bytes in dense float32, where #g is the number of genes. In our case, we have #g ≈ 60k genes, which corresponds to ~240 KB per cell. Accounting for a 2x headroom factor for intermediate computations, a 256 GB machine can hold roughly 500k cells in total across all workers. The exact breaking point will vary by the minibatch size, the number of workers and the workflow overhead, but this calculation provides a useful guide when choosing f.
>
> Q3:
> We do not expect a significant overhead simply from having many smaller AnnData objects. In practice, this is largely handled by the AnnData format itself and the AnnCollection abstraction. Because of that, both scDataset and AnnLoader would be affected in a similar way, so any performance impact should be comparable between the two rather than specific to one approach.
>
> Q4:
> In order to obtain good performance in real-world training, the minibatch entropy should be close to that of true random sampling. The key point is that this regime is easily achievable in practice. In particular, we find that relatively moderate values of the fetch factor already recover near-random diversity.
> As a practical guideline, we recommend setting the block size to at most half the fetch factor (i.e., b ≤ f/2), which typically preserves good batch entropy while maintaining efficiency. It is also worth noting that the Tahoe-100M setup represents a near worst-case scenario for block sampling, since cells are ordered by plate and thus highly correlated within blocks. Even in this setting, Figure 4 shows that b = f already preserves most minibatch diversity, so b = f/2 is a conservative recommendation that should work well across virtually all practical cases. We are happy to add this guideline explicitly in the paper if the reviewer thinks it would be a useful addition. See also answer to Q4 of reviewer a4Bm.

---

> > ### Author Rebuttal · Reviewer_Rntu · 2026-03-31
> >
> > I thank the authors for addressing my questions. I believe this paper is a useful contribution to training single-cell foundation models and also has applications in other domains. Therefore, I raise my score to 5.

---

> > > ### Author Response · Authors · 2026-04-02
> > >
> > > We thank the reviewer for their time, valuable feedback and positive assessment of the paper!

---

### Official Review · Reviewer_vDnG · 2026-03-13

**Soundness:** 2
**Presentation:** 3
**Significance:** 2
**Originality:** 2
**Overall Recommendation:** 4
**Confidence:** 3

**Summary:**

The paper proposes a new method for loading single cell omic data sets. Their method proposes a combination of block sampling and batched fetching which are optimised for disc IO overheads.

**Compliance With Llm Reviewing Policy:**

Affirmed.

**Final Justification:**

Building on the review.

The paper gives a useful tool which is well need by the biological (sc experiment) community.

The rebuttal (both to this review and others) has alleviated several of the concerns previous addressed. Consequently I have increased my score.

Some of the concepts batch-fetching are brought/motivated from existing tools in databases - the distinction about novelty is not clearly made. the authors have attempted to clarify their positions by citing an example of the hugging face challenge. However, existence of the challenge or difficulty does not imply the abscence of theses techniques but the abscence of their packaing in a more convinient form for the community (which is what this paper provides).

Lastly the I would have liked to see a more elaborate discussion of the application of this tool in areas other than single cell. Either that or additional optimizations made for single cell data (e.g. the effects of handling continuous integrations). The authors have attempted to cover this to some extent in the rebuttal.

**Key Questions For Authors:**

1. How does the framework allow for required stratified processing e.g. due to heterogenous experiments, batch effects?
2. How does the batched fetching used here compare to batched fetching widely used in database systems?
3. Batched sampling induces covariance within blocks. How does that variance scale with increases in block size? [is this taken care of empirically or are there theoretical justifications behind it]
4. What features of the framework make it geared specifically for single cell data as opposed to other data sources/types?

**Limitations:**

Technical limitations: to some extent
Societal: yes

**Strengths And Weaknesses:**

The authors identify a significant gap in utilising large single cell data sets. They propose a method which has the potential to be impactful to a large set of researchers.

The paper is well written with substantiation for claims. The background is well researched and explained. Code the code has also been released for comparison.

Despite the title the paper does not differentiate single cell data from other large volume data. Single cell data particularly is very high dimensional and often required stratified processing e.g. due to heterogenous experiments, batch effects. Discussing these factors would be essential for a paper tuned for single cell data.

While ideas like block sampling, buffer reshuffling, sorting indices, (batch fetching is prolific in database systems)  are not novel there packaging together is useful

Assumes physical contiguity reflects random biological mixing.

---

> ### Author Rebuttal · Authors · 2026-03-30
>
> We thank the reviewer for their time and feedback.
>
> Q1:
> We appreciate this important point. The scDataset framework supports stratified data loading through the BlockWeightedSampling and ClassBalancedSampling samplers. BlockWeightedSampling accepts arbitrary per-cell weights, allowing users to define any custom stratification scheme (e.g., correcting for batch effects or upsampling under-represented conditions). ClassBalancedSampling is a convenient wrapper where the user simply provides class labels, and weights are automatically derived to balance sampling across classes. We will add a sentence clarifying these capabilities in Section 3.3. Thanks for the suggestion!
>
> Q2:
> Thank you for the valuable observation. Batched fetching is indeed a well-established concept in database systems, where it primarily serves as a throughput optimization (loading data in bulk to amortize I/O costs). In our framework, batched fetching serves this purpose too, but more importantly it is the mechanism by which we recover randomness from block-structured sampling. By fetching more cells than strictly needed for a single training batch, we can perform in-memory shuffling across a larger pool, approximating true random sampling without incurring the random-access I/O penalties typical of cell-level reads. The key contribution is thus not batched fetching per se, but its integration with block sampling to jointly optimize throughput and sampling diversity.
>
> Q3:
> We assume the reviewer is referring to block sampling, please clarify if otherwise! Because cells within the same storage block tend to share experimental origin, block sampling preserves their correlation within minibatches, reducing the effective sample size from m to approximately m/b, as each block acts as a single independent unit. This effect is explicitly addressed in our theoretical analysis (Section 3.4), where we derive bounds on minibatch entropy as a function of block size b and fetch factor f. Crucially, increasing b reduces within-minibatch diversity and increases gradient variance, but batched fetching directly counteracts this: by drawing from f × m/b independently sampled blocks simultaneously, the effective diversity is restored. As f increases, the effective sample size approaches m, recovering behavior close to i.i.d. sampling as shown empirically in Figure 4.
>
> Q4:
> scDataset is intentionally designed as a general-purpose tool for fast quasi-random sampling from disk, and is not strictly limited to single-cell data. However, it is particularly well-suited to the single-cell omics ecosystem for two reasons: first, the scale of modern single-cell datasets (millions to hundreds of millions of cells) exposes the throughput limitations of existing solutions; second, by maintaining native compatibility with the AnnData format, scDataset integrates seamlessly with the broader single-cell toolchain (e.g., Scanpy, scvi-tools) without requiring data conversion. The framework's design choices are general enough to benefit any domain with large, disk-resident tabular or sparse data.

---

> > ### Author Rebuttal · Reviewer_vDnG · 2026-04-05
> >
> > thank you for the paper and rebuttal.
> >
> > 1. The assumption that *physical contiguity reflects meaningful structure* is still not completely addressed. Since much of the theoretical analysis and empirical validation depends on the Tahoe-100M layout, this remains unjustified. While the rebuttal discusses block-level correlation, it does not address what happens when storage layout does not align with experimental or biological grouping.
> >
> > 2a. The rebuttal gives a good conceptual distinction from database-style batched fetching. But could similar gains be observed from stronger existing batching and shuffle-buffer approaches under tuned settings?
> > 2b. Would the larger in memory shuffle required to reach this be a bottleneck? If yes, then what is the tradeoff (and how is it justified). If no, then please justify.
> >
> > 2. the single-cell-specific framing remains only partly supported. The rebuttal does not identify a methodological ingredient that is unique to SC biology - it rather acknowledges the generality. However the paper/rebuttal neither offers support (empirical) to support its generality nor does it highlight unique features to SC data. (Apart from integration with preexisting SC tools/pipelines).
> >
> > 3.The explicit statement that gradient variance scales approximately with $b/m$ via the reduction in effective sample size is a step in the correct direction. The connection between fetch factor $𝑓$ recovering an effective sample size closer to $m$ is also clear.
> > Despite that the argument is largely heuristic: entropy is used as a proxy for diversity, and effective sample size is used to reason about gradient variance, but there is no analysis (direct or empirical) of gradient variance or convergence behavior as a function of $b$ or $f$
> > 3a.To support the current statement Either explicitly frame this as an approximation grounded in standard SGD theory with clearer assumptions, or offer empirical measurements of gradient variance or training dynamics to validate the claim.

---

> > > ### Author Response · Authors · 2026-04-06
> > >
> > > We thank the reviewer for their follow-up questions!
> > > 1. When storage does not align with biological or experimental grouping, the dataset is effectively pre-shuffled by construction. The diversity machinery becomes less relevant and one can simply use streaming (see also Q3 answer to reviewer a4Bm).
> > >
> > > Overall, **scDataset provides throughput gains independently of the storage organization**; and when structure is present, it additionally recovers diversity.
> > >
> > > Importantly, **no assumption on physical contiguity is required for the validity of the method**; the plate-constant assumption reflects a worst-case analytical setting used to derive explicit entropy bounds.
> > >
> > > 2a/2b. Conventional shuffle buffers can in principle match the diversity of scDataset, but only by buffering a sufficiently large fraction of the dataset to overcome its underlying storage organization. On Tahoe-100M, this requires covering all 14 plates (~100M cells), which at ~240KB per cell amounts to ~24TB of RAM, which is impractical even on high-end machines. So yes, the in-memory buffer required to reach equivalent diversity is the bottleneck for conventional approaches. scDataset avoids this entirely by shuffling block indices rather than buffering data contiguously: **near-random diversity is achieved with a buffer of only f×m cells, keeping memory consumption modest and predictable**.
> > >
> > > 2. We focus on single-cell data because this is where the data loading problem is most acute in practice. In this setting, the bottleneck is substantial: on a NVIDIA DGX Station, training deep learning models on Tahoe-100M with **AnnLoader required approximately two months per epoch**. The Tahoe-100M competition on HuggingFace provides direct evidence of this limitation: every participating team resorted to pseudobulk aggregation because cell-level training on the full dataset was infeasible. The 100M-cell regime was, in practice, accessible only to groups with substantial dedicated infrastructure. Moreover, Tahoe-100M itself was released on HuggingFace alongside a conversion script back to AnnData, reflecting the fact that existing pipelines are tightly coupled to this format. In this context, **both high-throughput data access and seamless integration with existing tooling are critical, and scDataset directly addresses both**.
> > >
> > > At the same time, the method itself is general. It addresses a fundamental tension between data locality and sampling randomness, and the underlying mechanism, combining block-wise access with in-memory shuffling to trade off I/O efficiency and sampling diversity, applies to any large disk-resident dataset. We show this in our theoretical analysis, which **characterizes this tradeoff independently of the application domain, and empirically in Appendix D**, where we demonstrate similar throughput gains across other backends (e.g. HuggingFace Datasets and BioNeMo-SCDL (which uses NumPy memmap)). We will make this aspect more explicit in the revision and we thank the reviewer for their suggestion. See also answer 'About originality' to reviewer wKkT.
> > >
> > > 3a. We will explicitly frame the gradient variance connection as a formal approximation in Section 3.4. Under i.i.d. sampling, standard SGD theory (Bottou et al., 2018) gives $\text{Var}[\hat{g}] = \sigma^2/m$. Block sampling can be viewed as cluster sampling (Cochran, 1977): we draw $B = m/b$ blocks of size $b$ rather than $m$ individual cells. Under the plate-constant assumption, all $b$ cells in a block share the same gradient, so a minibatch contains only $B$ statistically independent gradient signals, giving $\text{Var}[\hat{g}] = \sigma^2/B = \sigma^2 b/m$. With fetch factor $f$, the buffer spans $f \cdot B$ independently sampled blocks; after in-buffer shuffling, each minibatch draws from this more diverse pool, reducing the effective cluster size per minibatch to approximately $b/f$ and giving $\text{Var}[\hat{g}] \approx \sigma^2 b/(fm)$. At $f = b$, the effective cluster size reaches 1 and i.i.d. variance is recovered, consistent with the empirical entropy results in Figure 4. We will state this explicitly as a heuristic approximation valid under the plate-constant model and standard SGD regularity conditions.
> > >
> > > Regarding empirical support: while we have not measured per-step gradient norm, the training loss curves are informative. **Streaming training exhibits spikes in training loss** that align precisely with plate transitions: when the stream shifts abruptly from one plate to another, it causes a sudden distribution shift in gradient direction. **BlockShuffling and random sampling produce smooth, well-behaved loss curves** throughout. The plots are available at https://postimg.cc/K44vqDgj and we will include them in the revised manuscript. This provides an empirical signature of elevated gradient variance under streaming, consistent with cluster-sampling effects.
> > >
> > > We thank the reviewer for the suggestions and hope that our rebuttal has addressed all the remaining questions.

---

### Decision · Program_Chairs · 2026-04-30

**Decision:**

Accept (regular)

**Comment:**

The paper develops a data loader for single-cell omics. A challenge with training models using mini-batch SGD on very large datasets involved in single-cell omics is that data with physical proximity is also similar, hence sampling contiguous batches leads to correlated samples. The paper develops a data loader using two standard techniques: Block Sampling which samples a contiguous block of datapoints, and Batched Fetching which samples multiple batches at once and mixes samples across them.

The paper is a good contribution from the engineering side. It develops a package which could be helpful for researchers working on single-cell omics. The reviewers all mostly appreciated it for this reason. However, the novelty is limited, block sampling is the standard streaming sampling scheme, and fetching multiple batches is standard in ML pipelines --- though these have not explicitly used to get data diversity while maintaining efficiency. The downstream impact on the learned model is evaluated in a limited way: the original paper only has result on a linear classifier. In the responses the authors discussed some preliminary results on MLPs. The gap between the standard streaming sampling (where one cycles across the data without worrying about data diversity) and the proposed approach in Fig. 5 is very small in 2 of the 4 examined setting (though the figure makes it appear larger due to the scale).

The reviewers did raise the above concerns, but their scores improved significantly during the authors response period (avg of 3.25 to 4.5). I personally am less satisfied that the concerns are addressed. There is a good applied contribution to the community working on  single-cell omics, and for this reason the paper can be accepted. However, I won't mind if the decision was bumped down and the paper was rejected.